# Comprehensive live-cell imaging analysis of cryptotanshinone and synergistic drug-screening effects in various human and canine cancer cell lines

Michael L. Bittner[1,2]☯, Rosana Lopes[1]☯*, Jianping Hua[1], Chao Sima[1], Aniruddha Datta[1]‡, Heather Wilson-Robles[ID][3]‡*

**1** Center for Bioinformatics and Genomic Systems Engineering, Texas A&M Engineering Experiment Station, Texas A&M University, College Station, TX, United States of America, **2** Translational Genomics Research Institute, Phoenix, AZ, United States of America, **3** College of Veterinary Medicine, Texas A&M University, College Station, TX, United States of America

☯ These authors contributed equally to this work.
‡ These authors also contributed equally to this work.
* rlopes@tamu.edu (RL); hwilson@cvm.tamu.edu (HWR)

**Data Availability Statement:** All relevant data are within the paper.

## Abstract

### Background

Several studies have highlighted both the extreme anticancer effects of Cryptotanshinone (CT), a Stat3 crippling component from *Salvia miltiorrhiza*, as well as other STAT3 inhibitors to fight cancer.

### Methods

Data presented in this experiment incorporates 2 years of *in vitro* studies applying a comprehensive live-cell drug-screening analysis of human and canine cancer cells exposed to CT at 20 μM concentration, as well as to other drug combinations. As previously observed in other studies, dogs are natural cancer models, given to their similarity in cancer genetics, epidemiology and disease progression compared to humans.

### Results

Results obtained from several types of human and canine cancer cells exposed to CT and varied drug combinations, verified CT efficacy at combating cancer by achieving an extremely high percentage of apoptosis within 24 hours of drug exposure.

### Conclusions

CT anticancer efficacy in various human and canine cancer cell lines denotes its ability to interact across different biological processes and cancer regulatory cell networks, driving inhibition of cancer cell survival.

**Funding:** This work was supported in part by the National Science Foundation under Grants ECCS-1609236 and ECCS- 1917166 and in part by the TEES-Agrilife Center for Bioinformatics and Genomic Systems Engineering (CBGSE) Startup Funds https://urldefense.com/v3/__https://www.nsf.gov/__;!!KwNVnqRv!VZFMWMeCrnsN302 XHXtCUEKedB81y4pQZnFrI8dWr7AJ_9d17M-QuzrX6HX9yYb6al8$ https://www.txgen.tamu.edu/research/cbgse/ The funders had no role in study design, data collection and analysis, decision to publish or preparation of the manuscript. The remaining funding for this work was provided by the Fred and Vola Palmer Chair for Comparative Oncology held by Dr. Robles.

**Competing interests:** The authors have declared no competing interests exist.

## Introduction

The elusive and frequently lethal behavior of cancer cells has led scientists to uncover differing biomolecular pathways and to test novel types of experimental methods in an attempt to cure cancer. Until now, several molecular cell approaches *in vitro* and *in vivo* have been utilized to determine the complex mechanisms by which cancer cells thrive and avoid cell death [1, 2]. The complexity of cancer has been highlighted over decades by treatment failure due to evasion of apoptosis, tumor microenvironment modulation, proliferation, metastatic behavior and drugs resistance [3, 4]. Despite the fact that human cancer rates are still the leading cause of death in many states in the U.S. Cancer is ranked as the second mortality cause in humans [5, 6]. While cancer treatment is far from its desired goal, progress has been achieved by discerning unique models of cancer epidemiology and behavior, increasing preventative care and improving diagnosis [7].

New approaches in treating cancer have evolved since the discovery of specific small molecules that selectively inhibit cell survival, proliferation, and migration in intricate pathways of mutated cancer cells [8–10]. In the last five years, a particularly useful type of anticancer therapy, that promotes the binding of signal transducer and activator of transcription 3 (STAT3) to Cryptotanshinone has been investigated. This binding tremendously reduces both the cellular production of STAT3, and the availability of the STAT3 dimers that drive production of various cellular oxygen and electron sources [11]. Signal transducer and activator of transcriptions (STATs), specially STAT3 and STAT5 are implicated in cancer cell survival, proliferation, migration and apoptosis inhibition [2, 12]. STAT3 signaling is derived from the Janus Kinase/STAT pathway and it is activated by the assembly of growth factors and cytokines in cell's surface receptors through tyrosine phosphorylation [8, 12]. Studies have shown that STAT3 nuclear translocation and continuous signaling activation of STAT3 proteins in tumors is achieved by cytoplasmic tyrosine kinase phosphorylation and STAT3 dimerization through Src homology 2 (SH2) domain interaction [8, 12, 13]. In healthy cells, STAT3 protein activation is not constant, it can occur from minutes to hours [12] and can be negatively regulated by phosphatases, inhibitors of cytokine signaling, and protein inhibitor of activated STAT (PIAS) [14]. In contrast, increased expression of STAT3 has been found in a variety of cancer cell lines and it is associated with poor disease prognosis and drug resistance [13, 15, 16]. In addition, post-translational modifications of STAT3 with phosphorylation at the serine 727 residue have been observed in the mitochondrial matrix and linked to cell energy metabolism regulation [14, 17]. Given the importance of STAT3's varied roles in cell metabolism and survival, STAT3 inhibitors have been considered as novel modes of cancer therapy [18, 19].

For thousands of years, a well-known Chinese plant, *Salvia miltiorrhiza* (Danshen), has been used to treat cardiovascular [20], gastrointestinal [21], circulatory [22] and neurological [22] diseases. Other benefits of *Salvia miltiorrhiza* include antibacterial [23], immunomodulatory [24] and antioxidant [25] activities. One of the main components of *Salvia miltiorrhiza*, Cryptotanshinone (CT), has been evaluated *in vitro* and *in vivo* as anticancer treatment in human tumor cell lines from different origins, such as, prostate [18, 26], breast [26–29] hepatic [30] and colorectal [18, 30–32], lung [33], pancreatic [34], kidney [35], glioma [36] and ovarian [37].

To date, several human studies have highlighted significant apoptotic effects of CT treatment in cancer cell lines using concentrations of 5–50 μM while sparing normal or healthy cell lines [27, 28, 32, 33–36]. Other effects of CT included cell cycle arrest at G1/G0 phase [30, 34, 38], and at G2/M phase [33, 39, 40], inhibition of Cyclin D1 protein expression in different types of cancer [18, 31, 33, 34, 37], and downregulation of matrix metallopeptidases (MMP-2/MMP-9), which are responsible for cell invasion and metastasis [41]. Human cancer cells

exposed to different concentrations of CT showed inhibition of major signaling pathways that are involved in multiple cellular processes such as, mammalian target of rapamycin (mTOR) in hepatic [30], gastric [30], pancreatic [34], soft sarcoma [38] and prostate [38] cancer. Janus kinase 2 (JAK2), phosphoinositide 3-kinase inhibitors/protein kinase B (PI3K/Akt) and extra-cellular related signal kinases (ERK) signaling was inhibited in pancreatic [34] and liver [1] cancer. The phosphorylation inhibition of STAT3 by CT treatment was reported in prostate [18], pancreatic [34], colorectal cancer [31], gastric [42], renal cell carcinoma [37], and malignant glioma [39].

In dogs and humans, increasing age is associated with higher cancer rates [6, 43, 44]. Recently, dogs have been considered as natural cancer models owing to their fast aging rate, selective breeding, similar environment exposure, and response to treatment comparable to humans [3, 45–47]. Based on genetic classification, epidemiology and disease progression, canine osteosarcoma has a high genetic similarity of that found in children [45, 48]. Osteosarcoma is one of the most common types of cancer in large breed dogs and ranked within the fourth most common malignancies [46]. Mutations on p53 tumor suppressor gene, loss of phosphatase and tensin homolog (PTEN), retinoblastoma (RB1) gene, and STAT3 [49] signaling alterations have been found in dogs and humans diagnosed with osteosarcoma [48, 50, 51]. Similar to osteosarcoma, malignant canine mammary tumors and human breast cancer share similar epidemiology and disease behavior showing breaking point cluster region 1 (BRCA1) mutations and susceptibility to steroid hormones [52]. Breast and mammary cancer gene expression analyses found that several genes including PI3K/Akt, KRAS, PTEN, WNT-beta catenin and MAPK pathways were altered in a similar mode in humans and dogs [53]. The mixed mammary cancer type is common in dogs and share similarities with the metaplastic breast cancer in humans [54]. Women6 and intact female dogs [43, 44] have a higher risk to develop mammary and skin cancer which are within the fifth most common types of cancer in both species. Canine malignant melanomas are commonly found in the oral mucosa, contrary to human melanoma that are most commonly observed in the skin. Although derived from different locations human and dog melanomas show aggressive behavior [55]. Mutations of PTEN and c-kit are found in both human and dog melanoma [56]. Interchangeable benefits can be achieved utilizing anticancer drugs approved by the U.S. Food and Drug Administration (FDA) to treat different types of cancer in humans and dogs [45].

This *in vitro* experiment was designed to establish the timely response of CT drug efficacy alone and combined with single drugs (MLN9708, SH4-54, HO-3867, lapatinib ditosylate, PX-478, paclitaxel, gefitinib, gemcitabine, capecitabine, metformin and cisplatin) in various human and canine cancer cell lines. The main goal in this study is to evaluate the dynamic response of cryptotanshinone alone and its synergistic effects with drug therapies using a live-cell imaging-based assay in human and canine cancer cell lines.

## Materials and methods

### Cell lines

All canine cancer cell lines including, ten canine osteosarcoma cell lines (ABRAMS, BKOS, CKOS, MCKOS, SKOS, UWOS-2, MC-KOSA, BW-KOSA, HEIDI, DAISY), six canine malignant melanoma (CML-1, CML6M, 17CM98, JONES, PARKS, KMSA) and two canine mammary cancer cell lines (REM, CMT12) were kindly provided by Dr. Heather M. Wilson-Robles, Small Animal Clinical Sciences Department, College of Veterinary Medicine, Texas A&M University. All fifty-one human cancer cell lines described in this study were obtain from the Translational Genomics Research Institute, Phoenix, Arizona.

## Drugs

All drugs used in this experiment were purchased from Selleckchem, Houston, TX-77014, USA and stored according to manufacturer instructions. Twelve drugs with different concentrations were used in drug-screening analyses including, Cryptotanshinone 20 μM (STAT3 inhibitor), HO-3867 10 μM (selective STAT3 inhibitors), SH-4-54 5 μM (STAT3/STAT5 inhibitor), Paclitaxel 0.1 μM (microtubule polymer stabilizer), Gefitinib 10 μM (EGFR inhibitor), PX-478 25 μM (hypoxia-inducible factor-1α inhibitor), Lapatinib Ditosylate 5 μM (EGFR and ErbB2 inhibitor), MLN9708 10 μM (chymotrypsin-like proteolytic β5 inhibitor), Metformin 10 mM (mTOR inhibitor), Capecitabine 50 μM (fluoropyrimidine carbamate), Cisplatin 40 μM, and Gemcitabine 2 μM (DNA synthesis inhibitors).

## Cell culture and sample preparation

Experiments were performed with various human and canine cancer cell lines from different origins seeded in triplicates over 24 columns in a microtiter plate and analyzed as a group. A total of seven microtiter plates containing fifty-one different human cancer cell lines were analyzed in this experiment. A plate composed by human pancreatic cancer cell lines included five pancreatic carcinomas (PANC1, MIA Paca-2, HS 766T, PANC1 TD2, P4057), two pancreatic adenocarcinomas (HPAF-II, AsPC1), and one pancreatic ductal adenocarcinoma (PL45). A second plate included four breast adenocarcinomas (MCF-7, MDA-MB-231, AU-565, MDA-MB-468), two breast carcinomas (MDA-MB-453, SUM159) and two breast ductal carcinoma cell lines (BT549, BT474). A third plate was formed by six human osteosarcoma cell lines: SaOS-2, 143B, G292, Hu03N1, HOS, HOS-MNNG, and one fibrosarcoma (HT1080) cell line. The fourth plate included eight human melanoma cancer cell lines (A375, A2058, SK-MEL-28, UACC-257, UACC-903, UACC-1093, UACC1308, UACC-2641) and, the fifth plate was composed by five human glioblastoma cell lines (GBM-12, U87MG, LN-18, LN-229, M059K) and two squamous cell carcinoma (CAL-27, SCC-9) cell lines. Additionally, a plate containing five lung cancer cell lines composed by two non-small cell lung cancer adenocarcinomas (H1975, H2073), a lung carcinoma cell line (A549), and two lung squamous cell carcinomas (SK-MES-1, SW900) was assembled. The last plate contained seven colon cancer cell lines composed by four colorectal adenocarcinomas (HT29, SW480, SW620, COLO205), two colorectal carcinomas (HCT116, LS513), one colon adenocarcinoma (DKS-8), and, one hepatocellular carcinoma cell line (Hep3B).

In addition, three microtiter plates containing eighteen different canine cancer cell lines were analyzed in this experiment. A plate composed by eight canine osteosarcoma cell lines included: ABRAMS, BKOS, CKOS, MCKOS, SKOS, UWOS-2, MC-KOSA and BW-KOSA. A second plate, was composed by six canine malignant melanoma cell lines (CML-1, CML6M, 17CM98, JONES, PARKS, KMSA), one mammary adenocarcinoma (REM) and one mammary carcinoma (CMT12) cell line. The remaining plate was formed by four canine osteosarcoma cell lines (ABRAMS, MCKOS, HEIDI and DAISY), the last two osteosarcoma cell lines had been recently collected from tumor site.

To optimize imaging analyses, a media with low levels of autofluorescence was prepared. The imaging media (IM) contained: 70% M-199 (11825015), 30% RPMI-1640 (11875085) supplemented with 10% FBS (16000044), 20 mM Hepes (15630080), 14 mM Glutamax (35050061), 7 mM of sodium pyruvate (11360070), 1% Penicillin-Streptomycin (15240062), 0.7 g of glucose (A2494001), 0.5 μM Vybrant® Dye-Cycle™ Violet Stain (V35003) and, Cell-Tox™ Green Cytotoxicity Assay (G8742) at 1:5,000 dilution. The Vybrant® Dye-Cycle™ Violet Stain is a live-cell permeable dye and produces blue fluorescence (< 437 nm) when bound to double-stranded DNA and stimulated by a violet excitation source (< 369 nm). In contrast,

the CellTox ™ Green Cytotoxicity Assay was incorporated in the media as indicative of cell death. This cyanine dye binds to DNA of cells that show lack of membrane integrity and produces a green fluorescence (< 509 nm) with a 488 nm excitation source.

Preceding the experiment, subconfluently grown cancer cells were dissociated using TrypLE^TM Select enzyme (12563029) and resuspended in IM media to establish the number of cells per milliliter in an automated cell counter (C10227). For each experiment, a 30 μl aliquot of each single cell line at density of 75,000 cells/well was delivered to a well in a 384-well microtiter plate (Greiner Bio-One 781091) pre-coated with 10 μg/ml of Rat Tail Collagen Type I (354249). Cells were cultured in the microtiter plate for 20 hours at 37°C in 5% $CO_2$ incubator prior imaging. The multiple cancer cell line samples were tested in triplicate wells, with a final volume of 60 μl per well in all microtiter plates.

### Drug screening and imaging

**Phase 1 –Single drug and drug combinations treatment.** After each microtiter plate was incubated for 20 hours at 37°C in 5% $CO_2$, triplicate wells seeded with different cancer cell lines labeled as untreated received an additional 30 ul aliquot of IM media per well and considered as controls. Two-time point images were recorded and saved before adding drugs to each microtiter plate well. Cancer cells were treated in three replicate wells with single drugs and drug combinations. The same amount of IM media (30 μl) containing two times the concentration of a single drug or drug combinations were added to each well in the microtiter plate rows and mixed by pipetting to all treated cancer cell lines. All experimental drugs were reconstituted in DMSO (D8418) at 10 to 50 mM stock concentration, except for Metformin 10 mM that was diluted in Dulbecco's Phosphate Buffered Saline (02-0119-0500-VWR). Cryptotanshinone 10 mM stocks were frozen at -20°C and prepared fresh every 14 days, other individual drug stock aliquots were stored at -20°C. For each plate, available drugs that have been previously used to treat each specific type of cancer were included to better evaluate the experimental drug responses to different cancer cell lines tested. The single drugs and their concentrations used in this experiment included: Cryptotanshinone 20 μM (STAT3 inhibitor), HO-3867 10 μM (selective STAT3 inhibitors), Paclitaxel 0.1 μM (microtubule polymer stabilizer), Gefitinib 10 μM (EGFR inhibitor), PX-478 25 μM (hypoxia-inducible factor-1α inhibitor), Lapatinib Ditosylate 5 μM (EGFR and ErbB2 inhibitor), MLN9708 10 μM (chymotrypsin-like proteolytic β5 inhibitor), Metformin 10 mM (mTOR inhibitor), Capecitabine 50 μM (fluoropyrimidine carbamate), Cisplatin 40 μM, and Gemcitabine 2 μM (DNA synthesis inhibitors). In addition, Cryptotanshinone at 20 μM concentration was paired with the following single drugs and their respective concentrations: HO-3867 10 μM, PX-478 25 μM were added to all types of cancer; Gefitinib 10 μM and Gemcitabine 2 μM (human pancreatic cancer), Paclitaxel 0.1 μM (breast and canine mammary cancer), Cisplatin 40 μM (human and canine osteosarcoma and human fibrosarcoma), Lapatinib Ditosylate 5 μM (human and canine melanoma, human glioblastoma), MLN9708 10 μM (human glioblastoma and human squamous cell carcinoma), Metformin 10 mM (human lung cancer), and Capecitabine 50 μM (human colorectal and human hepatocellular cancer). Other drug combinations included: MLN9708 10 μM plus SH-4-54 5 μM (STAT3/STAT5 inhibitor), Gemcitabine 2 μM plus HO-3867 10 μM, HO-3867 10 μM plus Placlitaxel 0.1 μM, HO-3867 10 μM plus Cisplatin 40 μM, HO-3867 plus Lapatinib Ditosylate 5 μM, HO-3867 10 μM plus Metformin 10 mM, and HO-3867 10 μM plus Capecitabine 50 μM. A combination of three drug replicates were selected for further testing based on the efficacy of above paired drugs to promote cancer cell death. The composition of drugs delivered in triplicates included: Cryptotanshinone at 20 μM concentration in all combinations plus possible groupings of two other drugs including, HO-

3867 10 µM, MLN9708 10 µM, Lapatinib Ditosylate 5 µM, SH-4-54 5 µM, Cisplatin 40 µM and PX-478 25 µM. A triplicate combination of HO-3867 10 µM plus MLN9708 10 µM plus SH-4-54 5 µM drugs was included to each plate. Untreated and treated cells were scanned every hour for 24 hours using an ImageXpress Micro XLS Widefield High-Content Analysis System (Molecular Devices) sampling at three different imaging sites within each well.

**Phase I1 –Assessment of cancer cell survival after drug treatments.** The first imaging data of human pancreatic, breast, osteosarcoma, melanoma, glioblastoma, lung, colon, fibro-sarcoma, squamous cell carcinoma and, hepatocellular carcinoma cancer cell lines cultured in seven individual plates was performed after 20 hours of incubation at 37˚C in 5% $CO_2$. Canine osteosarcoma, melanoma and mammary cancer cell lines cultured in two individual microtiter plates were kept at the same temperature and time conditions cited above. The ImageXpress Micro XLS Widefield High-Content Analysis System (Molecular Devices) was used to scan three different imaging sites within each well. Two-time points imaging for each cancer cell line run in triplicates were acquired and saved before the addition of drugs and considered as baseline. After the cells received different drugs and their combinations 24-time points imaging were obtained for each triplicate cancer cell line during a 24 hours period. After that, the plate was kept inside the incubator at 37˚C in 5% $CO_2$ for an additional 24 hours and a last time point was imaged and saved for each cancer cell line run in triplicates.

**Imaging processing.** All images were sent into an in-house pipeline developed in Matlab (Mathworks) using the SDC morphological toolbox [57]. After image segmentation on the nuclear channel and green fluorescent channels, a large number of measurements were collected for each individual cell line. Details on the image segmentation method were previously reported [58]. Briefly, there are two main parts to the image quantitative analysis. The first is the image processing where the fluorescent images are process and the transcriptional levels are extracted and quantified. For the second part, the extracted data are summarized into expression profiles where bar plots are generated from the transcriptional responses. Morphological features, nucleus size, nucleus mean intensity, and CellTox ™ Green mean intensity in nucleus were used for cell identification as well as cell death classification [59].

## Results

The *in vitro* efficacy of CT as anticancer agent has been investigated in cancer cell lines by our laboratory. The strength of our data is enhanced by the wide number of cell lines, different species examined, and the methodology applied with continuous live-cell monitoring in a timely mode improving results accuracy. Moreover, to our knowledge this is the first time CT anti-cancer effects have been evaluated in canine cancer cell lines in a systematic live-cell imaging-based assay. The main known effect of CT inhibiting cancer cell growth is linked to its capability of inhibiting STAT3 proteins [18, 34, 37]. Although STAT3 proteins are known to be ubiquitous in cells and found in the cytoplasm and mitochondrion, it is not known how the expression of STAT3 in different cancer cells can influence CT anticancer activity [11, 16]. STAT3 signaling activation stimulates transcription of different oncogenes that will perpetuate cell proliferation, cell survival and resistance to apoptosis [58]. Therefore, CT ability to bind to STAT3 proteins will impede STAT3 phosphorylation and subsequently STAT3 signaling activation [19]. Data obtained from this *in vitro* study using a live-cell imaging-based assay showed that CT at 20 µM concentration was able to induce cell death in various human and canine cancer cell lines. The single CT concentration of 20 µM utilized in this study was determined based on previous data evaluating cancer cells response to different CT concentrations. Results acquired in our laboratory from several live cancer-cell imaging data found that CT at 20 µM was effective to induce complete cell death in several cancer cell types (Fig 1). Moreover,

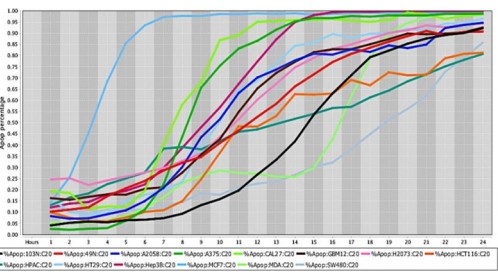

**Fig 1. Cell death response curve for a variety of human cancers.** Cell death response expressed in percentage of apoptosis in fourteen human cancer cell lines exposed to CT at 20 $\mu$M concentration. Cancer cell lines are represented by colored lines: colorectal cancer (103N- black, 49N- red, HCT116- orange, HT29- light blue, SW480- gray), melanoma (A2058- royal blue, A375- green), squamous cell carcinoma (CAL27- neon green), glioblastoma (GBM12- brown), lung adenocarcinoma (H2073- pink), pancreatic adenocarcinoma (HPAC- dark green), hepatocellular carcinoma (Hep3B- maroon), breast adenocarcinoma (MCF7- sky blue), breast carcinoma (MDA-MB-453- light green).

no significant changes in percentage of apoptosis in various human cancer cells examined was observed with CT concentrations above 20 μM (Fig 2). The cell death time in hours was attributed to each type of cancer when the percentage rate of cell death reached 80% in the majority of cancer cell lines exposed to CT (Table 1). The fastest rate that CT at 20μM induced cell death was observed in human fibrosarcoma and hepatocellular carcinoma (12 hours), followed by canine melanoma (13 hours) (Fig 3), human melanoma (16 hours), human glioblastoma, human squamous cell carcinoma (Fig 4A) and canine osteosarcoma (17 hours) (Figs 4B and 5), human osteosarcoma (18 hours), human breast (Fig 6) and human lung cancer (20 hours), human colorectal cancer (22 hours) (Fig 7), human pancreatic and canine mammary cancer (24 hours). In addition, CT at 20 μM concentration induced cell death in two recently collected canine osteosarcoma cell lines at 16 hours of incubation. Similar findings were observed in canine osteosarcoma cell lines with higher passage numbers.

The comprehensive and systematic drug-screening analyses of 51 human and 18 canine live-cancer cell imaging-based assay revealed the synergistic and dynamic response to CT treatment in several cancer cell lines. Results obtained from all human and canine cancer cell lines showed a remarkable decrease in cell death time in all types of cancer when selected drugs were combined to CT at 20 μM concentration (Table 1). The time in hours represent the point in time that most cancer cells achieved at least 80% percentage of apoptosis in several types of human and canine cancer exposed to a variety of drug concentrations. The drug-

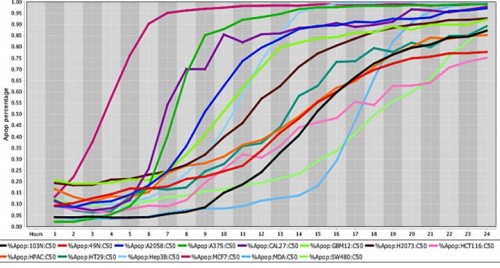

**Fig 2. Cell death response curve for a variety of human cancers.** Cell death response expressed in percentage of apoptosis in fourteen human cancer cell lines exposed to CT at 50 $\mu$M concentration. Cancer cell lines are represented by colored lines: colorectal cancer (103N- black, 49N- red, HCT116- pink, HT29- dark green, SW480- light green), melanoma (A2058- royal blue, A375- green), squamous cell carcinoma (CAL27- purple), glioblastoma (GBM12- neon green), lung adenocarcinoma (H2073- brown), pancreatic adenocarcinoma (HPAC- orange), hepatocellular carcinoma (Hep3B- light blue), breast adenocarcinoma (MCF7- maroon), breast carcinoma (MDA-MB-453- sky blue).

**Table 1. Time response in hours (h) to cell death induced by single drug and drug combinations screening in different human and canine cancer cell lines using a live-cell imaging-based assay.**

| Single drug and drug combinations / Cell lines | CT | CT + HO-3867 | CT + MLN9708 | CT + MLN9708 + SH-4-54 | CT + HO-3867 + SH-4-54 | CT + HO-3867 + Lapatinib Ditosylate | CT + HO-3867 + PX-478 /Gefitinib |
|---|---|---|---|---|---|---|---|
| Breast cancer | 20 h | 17 h | - | | - | - | 18 h (PX478) |
| Canine Mammary cancer | 24 h | 15 h | - | 16 h | - | 10 h | 13 h (PX478) |
| Human Colorectal cancer | 22 h | 20 h | - | 17 h | - | - | - |
| Human Fibrosarcoma | 12 h | - | - | 10 h | - | 11 h | - |
| Human Glioblastoma | 17 h | 16 h | 16 h | 13 h | 11 h | 14 h | - |
| Human Hepatocellular Carcinoma | 12 h | 8 h | - | 9 h | 8 h | 8 h | 9 h (PX478) |
| Human Lung cancer | 20 h | 13 h | - | 11 h | 13 h | 12 h | 11 h (PX478) |
| Human melanoma | 16 h | 11 h | - | 13 h | 11 h | 9 h | 13 h (PX478) |
| Canine melanoma | 13 h | 12 h | - | 12 h | 12 h | - | - |
| Human osteosarcoma | 18 h | 15 h | - | 10 h | 8 h | 11 h | 13 h (PX478) |
| Canine osteosarcoma | 17 h | 13 h | - | 11 h | 10 h | 11 h | 13 h (PX478) |
| Recently collected canine osteosarcoma cell lines | 16 h | - | - | - | 15 h | - | - |
| Human Pancreatic cancer | 24 h | 20 h | - | - | - | 19 h | 14h(Gefitinib) 19 h (PX478) |
| Human Squamous Cell Carcinoma | 17 h | - | 12 h | 12 h | - | - | - |

CT- Cryptotanshinone (20 μM), HO-3867 (10 μM), MLN9708 (10 μM), SH-4-54 (5 μM), Lapatinib Ditosylate (5 μM), PX-478 (25 μM), and Gefitinib (10 μM)

screening analyses shown in this study suggests that recently collected canine osteosarcoma cell lines required a longer time of exposure to selected drug combinations to induce cell death response.

## Drug efficacy measured by percentage of apoptosis.

The percentage values of cell death were determined using a live-cell imaging-based dynamic response trajectory [59]. This live-cancer cell imaging technology provides hourly measurements of continuous imaging sets from each cancer cell, seeded in a 384-well microtiter plate, to closely monitor the characteristic changes in cell morphology in a timeline mode. Moreover, to accurately discriminate between live and dead cells, live-cell fluorescent dyes were incorporated in the growth media to ultimately determine the percentage of cell death enhancing our capabilities. Drug efficacy in this study was measured by the power of CT and selected drug

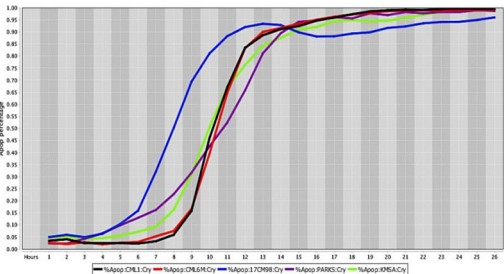

**Fig 3. Cell death response curve for canine melanoma.** Cell death response expressed in percentage of apoptosis in canine melanoma cell lines exposed to CT at 20 μM concentration (CML1- black line, CML6M- red line,17CM98- blue line, PARKS- purple line, KMSA- neon green line).

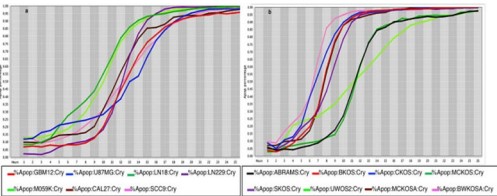

**Fig 4. Cell death response curves for glioblastoma and canine osteosarcoma.** 4a –Cell death response expressed in percentage of apoptosis in human glioblastoma (GBM12- red line, U87MG- blue line, LN18- dark green line, LN229- purple line, M059K - neon green line) and squamous cell carcinoma cell lines (CAL27- black line, SCC9- pink line) exposed to CT at 20 $\mu$M concentration. 4b –Cell death response expressed in percentage of apoptosis in canine osteosarcoma cell lines exposed to CT at 20 $\mu$M concentration (ABRAMS- black line, BKOS- red line, CKOS- blue line, MCKOS- dark green line, SKOS- purple line, UWOS-2- neon green line, MCKOSA- brown line, BWKOSA- pink line).

combinations to induce cell death. The ability to monitor different cancer cell types every hour for 24 hours or more, emphasizes the strength of utilizing a drug-screening methodology using a live-cell imaging-based assay. CT and selected drug combinations showed a percentage of apoptosis between 96% to 100% in human osteosarcoma (Fig 8A), and 98% to 100% in canine osteosarcoma cell lines (Fig 8B). In both species, treated cells reached the maximum percentage of apoptosis in an average of 12.5 hours. However, cells from recently collected canine osteosarcoma, treated with CT and selected drug combinations displayed an average of 15.5 hours to reach maximum percentage of apoptosis. A rate of 92% to 100% percentage of apoptosis in glioblastoma cell lines was observed within an average of 14.5 hours of incubation with CT and selected drug combinations (Fig 9). Human melanoma cells treated with CT and selected drug combinations showed 98% to 100% apoptosis, and canine melanoma cell lines had 92% to 100% percentage of apoptosis. In both species melanoma cell death was observed at an average of 12.25 hours of incubation. However, canine melanoma cancer cells did not show similar cell death response to a few drug combinations compared to human melanoma (Table 1). Human lung cancer showed an average of 13.3 hours of incubation to achieve 98% to 100% percentage of apoptosis exposed to CT and selected drug combinations (Fig 10). A higher time with an average of 19.7 hours of exposure to CT and selected drug combinations was required for human colon cancer cells to achieve a maximum percentage of apoptosis of 90% to 100%. Human breast cancer and canine mammary cancer showed 91% to 100% maximum percentage of apoptosis in response to CT and selected drug combinations at an average of 18.3 and 15.6 hours, respectively. Human pancreatic cancer showed a 90% to 100% maximum percentage of apoptosis in response to CT and selected drug combinations at an average of 19.2 hours of exposure to drugs (Fig 11). The single cell lines human fibrosarcoma, hepatocellular carcinoma and squamous cell carcinoma examined in this study displayed a 98% to

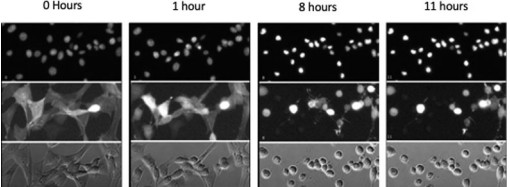

**Fig 5. Cell death images from canine osteosarcoma cell line UWOS-2.** UWOS-2 was exposed to CT at 20 $\mu$M concentration for 24 hours. Serial images of the cells over the first 12 hours of incubation are presented here. The top two panels include fluorescent images of the cells (top panel nuclei are stained; middle panel actin fibers are stained) and the bottom panel includes the images of unstained cells.

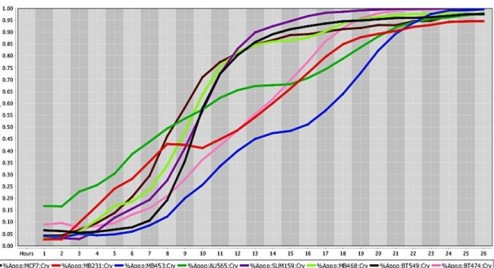

**Fig 6. Cell death response curves for human breast cancer.** Cell death response expressed in percentage of apoptosis in human breast cancer cell lines exposed to CT at 20 $\mu$M concentration (MCF7- black line, MDA-MB-231- red line, MDA-MB-453- blue line, AU565- dark green line, SUM159- purple line, MDA- MB-468- neon green light, BT549- brown line, BT474- pink line).

100% percentage of apoptosis in response to CT and selected drug combinations at an average of 11, 9, and 13.7 hours, respectively.

Cancer cells response to drug-screening was assessed by changes in cell morphology combined to intensity of fluorescent dyes and compared to untreated cancer cell lines. The cell nucleus size and nucleus mean intensity analyzed using a blue live-cell dye in untreated cancer cells revealed a homogeneously stained nucleus. In contrast, cells exposed to CT and drug combinations displayed increasing nucleus condensation in a timeline fashion. Cell death was determined by measuring CellTox$^{TM}$ Green (CTG) mean intensity in the cell nucleus as result of cancer cell membrane rupture and binding of CTG dye to DNA. Increased CTG nucleus intensity and pyknosis is a typical characteristic of cell death.

## Discussion

Although several *in vitro* studies have demonstrated the anticancer properties of CT in human cancer cells, we present here a reproducible and comprehensive live-cancer cell imaging-based assay that investigates drug efficacy in different species. The first objective in the present study, was to determine the response of CT using an *in vitro* live-cell imaging-based assay in human and canine cancer cell lines. The second objective, was to investigate the synergistic effects of CT and drug combinations utilizing the same methodology in human and canine cancer cell lines. This is the first time CT anticancer effects have been evaluated in canine cancer cells in a systematic live-cell imaging-based assay. Findings emphasize that CT at 20 μM concentration can induce cell death in the majority of human and canine cancer cells examined within 24 hours of drug exposure. Moreover, CT in combination with other single drugs including, HO-

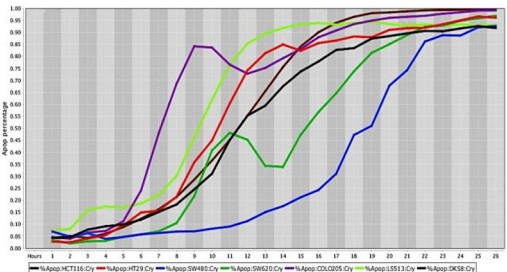

**Fig 7. Cell death response curves for human colorectal cancer.** Cell death response expressed in percentage of apoptosis in human colorectal cancer cell lines exposed to CT at 20 $\mu$M concentration (HCT116- black line, HT29- red line, SW480- blue line, SW620- dark green line, COLO205—purple line, LS513- neon green line, DSK8- brown line).

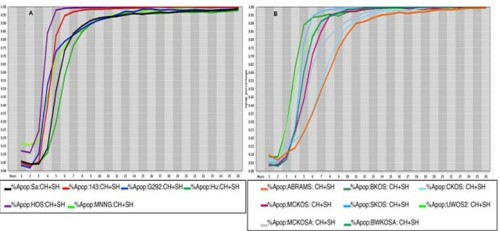

**Fig 8. Cell death response curves in human and canine osteosarcoma.** 8a Cell death response expressed in percentage of apoptosis in human osteosarcoma cell lines (SaOS- black line, 143B- red line, G292- blue line, Hu03N1- dark green line, HOS- purple line, HOS-MNNG- neon green line) exposed to CT at 20 $\mu$M, HO-3867 at 10 $\mu$M, and SH-4-54 at 5 $\mu$M concentration. 8b Cell death response expressed in percentage of apoptosis in canine osteosarcoma cell lines (ABRAMS- orange line, BKOS- dark green line, CKOS- light blue line, MCKOS- maroon line, SKOS- sky blue line, UWOS-2- neon green line, MCKOSA- gray line, BWKOSA- light green line) exposed to CT at 20 $\mu$M, HO-3867 at 10 $\mu$M, and SH-4-54 at 5 $\mu$M concentration.

3867, MLN9708 or paired drugs such as, MLN9708 plus SH-4-54, HO-3867 plus SH-4-54, HO-3867 plus Lapatinib, and HO-3867 plus Gefitinib or PX-478 produced synergistic effects observed by further decreasing cell death time in different types of cancer.

The efforts being applied in this study are aimed at ceasing to continue the enormous number of struggles to build up cancer treatments based on simply attempting testing of endless numbers of chemicals and proteins, to hopefully find sets which can be applied to a cancer patient to sufficiently eliminate the cancer's disruption of the patient's survival. A very clear demonstration of how a biological logician could concentrate on discerning what and how cellular components keep shifting in response to an organism's continuous management of its bodily components was described by Waddington [60]. In this line of thought, one of the most important issues in the fight against cancer is blocking the components required for cancer cell life. As cancer cells can exhibit a reservoir of highly active STAT3, it seems as though STAT3 can act as the essential director of the required fueling operations for cells that are carrying out extreme survival activities. A very direct way to test this is to apply a drug, or drugs, that can inactivate STAT3's actions, and determine rates of death of those cancer cells reliant on high levels of STAT3. While it has taken considerable time for efforts to design effective STAT3 inhibitors, a small number of effective inhibitors have been produced in the last seven years. STAT3 protein can become very effective in sponsoring the production and canalization of its own two protein domain combined form, as well as the combined forms of other proteins that also support cellular oxygenation. Alternatively, if STAT3 production is not high, what STAT3 is available may be further less active due to formation of a STAT3 and CT joint

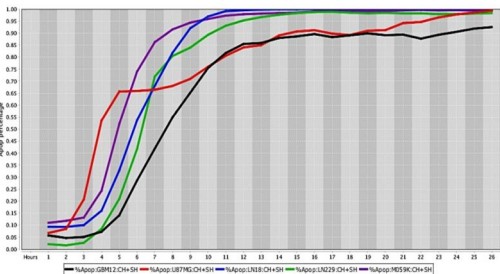

**Fig 9. Cell death response curves for human glioblastomas.** Cell death response expressed in percentage of apoptosis in human glioblastoma cell lines (GBM12- black line, U87MG- red line, LN18- blue line, LN229- green line, M059K- purple line) exposed to CT at 20 $\mu$M, HO-3867 at 10 $\mu$M, and SH-4-54 at 5 $\mu$M concentration.

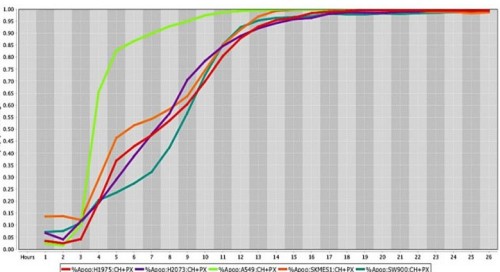

**Fig 10. Cell death response curves for human lung tumors.** Cell death response expressed in percentage of apoptosis in human lung cancer cell lines (H1975- red line, H2073- purple line, A549- neon green line, SKMES1- orange line, SW900- dark green line exposed to CT at 20 $\mu$M, HO-3867 at 10 $\mu$M, and PX-478 at 25 $\mu$M concentration.

product. CT is a protein that Chinese researchers extracted from plant root systems which will readily bind to STAT3. This will block the formation of other oxygenating products based on STAT3:CT providing another way of eradicating cancer cells.

Based on our results, four types of human cancer cell lines exposed to CT treatment including, pancreatic, breast, lung and colorectal cancer cells showed an increased time to attain cell death. Similar observations were found in canine mammary cancer, and recently collected canine osteosarcoma cells from naturally occurring disease. Recent genetic classification of various canine tumors [48, 49, 52, 54, 56] show that similar to humans, cancers in dogs are also known to have an invasive behavior and show resistance to drugs. In the present study, canine osteosarcoma, human osteosarcoma and glioblastoma cell lines, exhibited similar response to single CT treatment and cancer cell death was further enhanced by introducing another STAT3/5 inhibitor, a proteasome inhibitor drug, and an EGFR/ErbB2 inhibitor. Up-regulated STAT3 signaling has been previously found in human and canine osteosarcoma cell lines, and the effects of a small molecule inhibitor that prevented STAT3/DNA interaction, and further STAT3 cancer cell's transcription has been documented [49]. Similarly, it has been observed that glioma-initiating cells proliferation, and tumor growth are intrinsically linked to STAT3 up-regulation [17]. A significant decrease in cell death time to 13 hours in all human glioblastoma cells analyzed *in vitro* was observed in this study using a combination of CT, a proteasome inhibitor drug, and a STAT3/5 inhibitor. Abnormal proteasome activation has been associated to excess of unfolded protein response and subsequently endoplasmic reticulum distress [61]. A further increase in glioblastoma cell's death was observed when CT, a selective STAT3 inhibitor, and other STAT3/5 inhibitor were combined reducing the cell

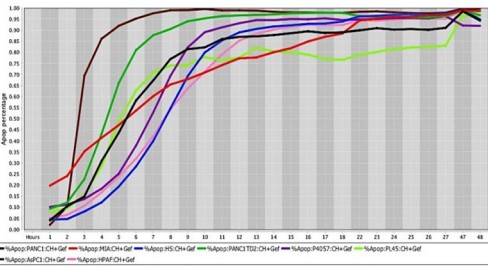

**Fig 11. Cell death response curves for pancreatic cancer.** Cell death response expressed in percentage of apoptosis in human pancreatic cancer cell lines (PANC1- black line, MIA Paca-2- red line, HS766T- blue line, PANC1 TD2- dark green line, P4057- purple line, PL45- neon green line, AsPC1—brown line, HPAF- pink line) exposed to CT at 20 $\mu$M, HO-3867 at 10 $\mu$M, and Gefitinib at 10 $\mu$M concentration.

death time to 11 hours. These results confirm previous reports [17, 38] suggesting that activated STAT3 response is one of altered signaling pathways involved in the proliferation and survival of tumor cells in human gliomas. In general, several factors influence treatment of brain tumors *in vivo*, starting with the blood brain barrier that contributes to decreased drug delivery efficacy needed to induce cancer cell death. Nonetheless, significant tumor reduction size and increased survival time in mouse presenting human glioma orthotopic xenograft was previously observed after CT treatment [39]. Therefore, human and canine osteosarcoma, and human glioma cells showed increased cell death response when exposed to STAT3 inhibitor drugs.

Hypoxia inducible factor 1- alpha (HIF-1α) up-regulation is involved in cancer cell proliferation, metastasis, resistance to therapy and worse prognosis [61]. HIF-1α is known to induce low oxygen levels found in the tumor microenvironment, mainly in solid tumors. In addition, overexpression of (HIF-1α), has been found to be regulated by STAT3 signaling [15]. Several cancer cell lines examined in this study, including human hepatocellular carcinoma, lung cancer, breast and canine mammary cancer, human and canine osteosarcoma, human melanoma and pancreatic cancer showed a consistent and quick cell death response when a HIF-1α was combined to CT and a selective STAT3 inhibitor. Moreover, results in this experiment show remarkable synergistic effects observed in pancreatic cancer cells treated with selective STAT3 inhibitors, an EGFR/ErbB2 inhibitor, and a HIF-1α inhibitor. Pancreatic cancer treated with CT, a selective STAT3 inhibitor, and Gefitinib (EGFR inhibitor) also showed encouraging results with the lowest time capable of inducing cell death in all eight pancreatic cancer cells examined. Similar synergistic effects *in vitro* and *in vivo* have been reported in ovarian cancer with a combination of Gefitinib and a JAK/STAT3 inhibitor drug [62].

Findings obtained in the present study revealed that cell death in seven human colorectal cancer, and eight human breast cancer cell lines exposed to CT at 20 μM concentration, occurred at 22, and 20 hours, respectively. In contrary, a previous report indicated that CT at a 20 μM concentration induced cell death to human breast carcinoma (MCF-7) cells within 72 hours of drug exposure [26]. The reason for several hours of delay in breast cancer cell death in the prior study could be attributed to different methodology applied, drug composition, and drug delivery procedures. Synergistic effects showing a decrease from 20 to 17 hours in cell death time was observed in all human breast cancer cell lines when a selective STAT3 inhibitor was combined to CT. Similar to previous data in human and canine cancer cell lines, adding a selective STAT3 inhibitor reduced colorectal cancer cell's death from 22 to 20 hours. Previous *in vitro* study in colorectal cancer cells treated with CT showed decreased cell invasion, and *in vivo* CT treatment decreased colorectal cancer tumor size, promoted downregulation of PI3K/Akt/mTOR signaling and inhibition of HIF-1α factor [63]. It has been observed that STAT3 signaling is responsible for up-regulation of HIF-1α, and both activated pathways lead to cell proliferation, metastasis, drug resistance and poor prognosis [15, 64]. It is also well known that CT induce STAT3 inhibition [31, 38] and several reports have demonstrated CT inhibitory effects on major cancer signaling pathways in pancreatic cancer [34], and cholangiocarcinoma[1] cells. Moreover, inhibition of PI3K/Akt/mTOR pathway has also been observed in ER-negative breast cancer cells (Bcap37) treated with CT at low concentration [29] and ERα-positive breast cancer cells [28]. Results in the present study from *in vitro* human pancreatic, glioblastoma, colorectal cancer, osteosarcoma and canine osteosarcoma cells treated with CT at 20 μM concentration, and drug combinations corroborate findings from various reports showing CT anticancer properties. In the same way, our results confirm that canine mammary cancer cells showed an equivalent response to CT treatment, and the addition of a selective STAT3 inhibitor enhanced cell death emphasizing the biological and genetic resemblance to human breast cancer [53].

CT anticancer effects have also been analyzed *in vitro* and *in vivo* in human non-small cell lung cancer (NSCLC) at different CT concentrations (10 to 40 µg/ml) and found that exposure to higher CT concentrations showed reduction in lung cancer size and cell's survival time [65]. Results in the present study confirm CT anticancer effects at 20 µM concentration and cell death was observed after 20 hours of drug exposure in human lung cancer cell lines. Related findings from *in vitro* and *in vivo* studies in lung adenocarcinoma cells showed cancer cell growth inhibition and decrease in tumor xenograft size by CT at different concentrations [33]. In addition, similar to our observations, the latest study reported no significant differences in cancer cell growth inhibition increasing CT concentrations above 20 µM concentration. Human and dogs with melanoma have shown alterations in similar signaling pathways including RAS/MAPK, PI3K/Akt, and mTOR [53]. Although RAS mutations in canine melanoma have not been frequently found in spontaneous disease [66], human and canine melanoma cells in the present study showed a similar response to CT and drug combinations. Moreover, recent *in vitro* report in human melanoma cells have shown that CT concentrations of 1.25 to 20 µM were capable to induce apoptosis [40].

Though CT demonstrated efficacy as a single agent against a variety of cancer cell lines in this study, the enhanced effects that CT had when used in combination cannot be overlooked. CT has been shown to restore sensitivity to TRAIL (tumor necrosis factor-related apoptosis inducing ligand) in a variety of cancer cells by inducing TRAIL receptor 2 (DR5) expression and TRAIL induced apoptosis in one study [67]. Another study demonstrated the ability of CT to potentiate the antitumor effects of doxorubicin on gastric cancer cells through down regulation of STAT3 target genes such as Bcl-xL, Mcl-1, surviving and XIAP [38]. CT has also been shown to inhibit P-glycoprotein (P-gp) mediated drug efflux from cells counteracting a major mechanism of resistance to anti-cancer drugs in colon cancer cells. It is presumed that CT down regulates P-gp mRNA and protein levels leading to decreased P-gp ATPase activity [68]. These studies add credence to the use of CT in combination with other traditional chemotherapy agents for the treatment of cancer and may explain, in part, the enhanced cell killing seen with a variety of drug combinations reported here.

Little is known about CT absorption, metabolism and its bioavailability in humans and animals [69, 70]. A low CT concentration percentage of 6.7 and 11.1 have been measured after oral dose in rats and dog's plasma samples, respectively. However, detection of CT and metabolites in concentrations of 32 to 191 times higher have been found mainly in stomach, intestine, lung, and liver after oral treatment with CT formulated in an inclusion-complex form [69]. Higher concentrations of CT and metabolites after intravenous injection also was observed in lungs and liver [69]. The increased CT concentrations found in selected organs including, stomach, intestine, lung and liver indicate that these organs can be the first-line targets to test CT anticancer properties in naturally occurring disease in dogs. Evaluation of CT bioavailability and CT plasma concentrations in healthy human subjects was assessed using a micronized granule powder technology leading to higher CT plasma concentrations compared to traditional CT plant extraction [70]. In addition, low cytotoxicity effects were found in healthy human cell lines exposed to CT [71].

Data presented in this study emphasizes the importance of *in vitro* experiments to identify the response from single drugs and drug combinations according to each cancer cell type. Cancer complexity can be addressed by utilizing synergistic drug combinations leading to a speedy cell death rate minimizing cancer cells' capabilities of invasion, metastasis and stemness behavior. Nonetheless, *in vivo* testing is invaluable to determine dosage, drug efficacy and possible side effects triggered by anticancer drugs candidates. The long history of benefits from *Salvia miltiorrhiza* associated with CT, countless *in vitro* and *in vivo* reports of anticancer efficacy, and inhibition of several major cancer signaling pathways, suggests that CT has great potential as anticancer drug.

## Author Contributions

**Conceptualization:** Michael L. Bittner.

**Data curation:** Michael L. Bittner, Rosana Lopes, Jianping Hua, Chao Sima, Heather Wilson-Robles.

**Formal analysis:** Michael L. Bittner, Rosana Lopes, Jianping Hua, Chao Sima.

**Investigation:** Rosana Lopes.

**Methodology:** Michael L. Bittner, Rosana Lopes, Aniruddha Datta, Heather Wilson-Robles.

**Project administration:** Michael L. Bittner.

**Resources:** Aniruddha Datta, Heather Wilson-Robles.

**Supervision:** Michael L. Bittner.

**Validation:** Rosana Lopes.

**Writing – original draft:** Michael L. Bittner, Rosana Lopes, Jianping Hua, Chao Sima, Aniruddha Datta, Heather Wilson-Robles.

**Writing – review & editing:** Michael L. Bittner, Rosana Lopes, Jianping Hua, Chao Sima, Aniruddha Datta, Heather Wilson-Robles.

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
