## [Decision Letter · Decision Letter 0]

26 Aug 2020

PONE-D-20-19784

Comprehensive Live-cell Imaging Analysis of Cryptotanshinone and Synergistic Drug-Screening Effects in Various Human and Canine Cancer Cell Lines

PLOS ONE

Dear Dr. Wilson-Robles,

Thank you for submitting your manuscript to PLOS ONE. After careful consideration, we feel that it has merit but does not fully meet PLOS ONE’s publication criteria as it currently stands. Therefore, we invite you to submit a revised version of the manuscript that addresses the points raised during the review process.

We look forward to receiving your revised manuscript.

Kind regards,

Chakrabhavi Dhananjaya Mohan, Ph.D

Academic Editor

PLOS ONE

Journal Requirements:

2. Please provide additional information about each of the cell lines used in this work, including the culture conditions and any quality control testing procedures (authentication, characterisation, and mycoplasma testing). For more information, please see http://journals.plos.org/plosone/s/submission-guidelines#loc-cell-lines.

3. In the Methods section, please provide the product number and any lot numbers of the inhibitors purchased from Selleckchem for your study.

4. To comply with PLOS ONE submission guidelines, in your Methods section, please provide additional information regarding your statistical analyses. For more information on PLOS ONE's expectations for statistical reporting, please see https://journals.plos.org/plosone/s/submission-guidelines.#loc-statistical-reporting.

Reviewers' comments:

Reviewer's Responses to Questions

**Comments to the Author**

1. Is the manuscript technically sound, and do the data support the conclusions?

Reviewer #1: Yes

Reviewer #2: Partly

2. Has the statistical analysis been performed appropriately and rigorously? 

Reviewer #1: Yes

Reviewer #2: I Don't Know

3. Have the authors made all data underlying the findings in their manuscript fully available?

Reviewer #1: Yes

Reviewer #2: Yes

4. Is the manuscript presented in an intelligible fashion and written in standard English?

Reviewer #1: Yes

Reviewer #2: Yes

5. Review Comments to the Author

Reviewer #1: The research article entitled “ Comprehensive Live-cell Imaging Analysis of Cryptotanshinone and Synergistic Drug2 Screening Effects in Various Human and Canine Cancer Cell Lines” by Bittner etal, have described the exposure of varied drug combination against human and canine cancer cells and highlighted the results of CT efficacy at combating cancer using comprehensive live-cell drug-screening analysis. The experiments are well designed and detail investigation/interpretations of the results have been carried out from the authors. The results are statistically validated and are in the interest of researchers with significant data which will be useful for cancer therapeutics. I feel the article can be accepted for publication. However minor spelling mistakes have been noticed which can be fixed by the authors.

Reviewer #2: This study reported the anticancer activity of cryptotanshinone against a number of human and canine cancer cell lines using a comprehensive live-cell imaging analysis. These findings are interesting, and provide useful information regarding the potential application of this natural product in treating cancer. The following concerns need to be addressed.

1. The authors may consider including several normal cells in the imaging analysis in order to show the selective cytotoxicity of cryptotanshinone.

2. Since this is a cell imaging analysis, it would be better to show some representative images of cells after treatment with cryptotanshinone, in particular a time-dependent response.

3. Regarding the anticancer activity of drug combination with cryptotanshinone, the potential application of cryptotanshinone in reversing multidrug resistance of cancer cells needs to be discussed.

4. A detailed method regarding quantitative analysis of the images needs to be described.

6. PLOS authors have the option to publish the peer review history of their article (what does this mean?). If published, this will include your full peer review and any attached files.

Reviewer #1: No

Reviewer #2: No

---

## [Author Response · Author response to Decision Letter 0]

10 Nov 2020

PONE-D-20-19784

Comprehensive Live-cell Imaging Analysis of Cryptotanshinone and Synergistic Drug-Screening Effects in Various Human and Canine Cancer Cell Lines

PLOS ONE

Dear Chakrabhavi Dhananjaya Mohan, Ph.D,

We thank you very much for the opportunity to revise this manuscript and the reviewers’ thoughtful comments. We have attempted to address all of the comments as outlined below in the updated manuscript. A response to each review is included below in italics. 

Reviewer #1: The research article entitled “ Comprehensive Live-cell Imaging Analysis of Cryptotanshinone and Synergistic Drug2 Screening Effects in Various Human and Canine Cancer Cell Lines” by Bittner etal, have described the exposure of varied drug combination against human and canine cancer cells and highlighted the results of CT efficacy at combating cancer using comprehensive live-cell drug-screening analysis. The experiments are well designed and detail investigation/interpretations of the results have been carried out from the authors. The results are statistically validated and are in the interest of researchers with significant data which will be useful for cancer therapeutics. I feel the article can be accepted for publication. However minor spelling mistakes have been noticed which can be fixed by the authors.

The manuscript has been reviewed and any typos that we identified have been fixed. 

Reviewer #2: This study reported the anticancer activity of cryptotanshinone against a number of human and canine cancer cell lines using a comprehensive live-cell imaging analysis. These findings are interesting, and provide useful information regarding the potential application of this natural product in treating cancer. The following concerns need to be addressed.

1. The authors may consider including several normal cells in the imaging analysis in order to show the selective cytotoxicity of cryptotanshinone.

Though normal cell lines were not tested in this study, we have provided the following references in the introduction as well as some additional language that support the use of this drug against cancer cell lines while sparing normal/healthy cell lines. 

2. Since this is a cell imaging analysis, it would be better to show some representative images of cells after treatment with cryptotanshinone, in particular a time-dependent response.

An additional figure (figure 5) has been added to the manuscript to show the effects of CT on the canine OS cell lines UWOS2 over 12 hours. 

3. Regarding the anticancer activity of drug combination with cryptotanshinone, the potential application of cryptotanshinone in reversing multidrug resistance of cancer cells needs to be discussed.

A paragraph discussing this has been added to the discussion. 

4. A detailed method regarding quantitative analysis of the images needs to be described.

The following reference and more detail have been added to the methods section describing in detail the quantitative analysis of the images. 

Hua, Jianping, Chao Sima, Milana Cypert, Gerald Gooden, Sonsoles Shack, Lalitamba Alla, Edward Smith, Jeffrey M. Trent, Edward R. Dougherty, and Michael L. Bittner. "Tracking transcriptional activities with high-content epifluorescent imaging." Journal of biomedical optics 17, no. 4 (2012): 046008.

Sincerely,

Heather Wils

---

## [Decision Letter · Decision Letter 1]

1 Dec 2020

Comprehensive Live-cell Imaging Analysis of Cryptotanshinone and Synergistic Drug-Screening Effects in Various Human and Canine Cancer Cell Lines

PONE-D-20-19784R1

Dear Dr. Wilson-Robles,

We’re pleased to inform you that your manuscript has been judged scientifically suitable for publication and will be formally accepted for publication once it meets all outstanding technical requirements.

Kind regards,

Chakrabhavi Dhananjaya Mohan, Ph.D

Academic Editor

PLOS ONE

Additional Editor Comments (optional):

Reviewers' comments:

Reviewer's Responses to Questions

**Comments to the Author**

1. If the authors have adequately addressed your comments raised in a previous round of review and you feel that this manuscript is now acceptable for publication, you may indicate that here to bypass the “Comments to the Author” section, enter your conflict of interest statement in the “Confidential to Editor” section, and submit your "Accept" recommendation.

Reviewer #1: All comments have been addressed

Reviewer #2: All comments have been addressed

2. Is the manuscript technically sound, and do the data support the conclusions?

Reviewer #1: Yes

Reviewer #2: Yes

3. Has the statistical analysis been performed appropriately and rigorously? 

Reviewer #1: Yes

Reviewer #2: Yes

4. Have the authors made all data underlying the findings in their manuscript fully available?

Reviewer #1: Yes

Reviewer #2: Yes

5. Is the manuscript presented in an intelligible fashion and written in standard English?

Reviewer #1: Yes

Reviewer #2: Yes

6. Review Comments to the Author

Reviewer #1: The authors have revised and addresses all the comments. The article can be accepted for publication

Reviewer #2: (No Response)

7. PLOS authors have the option to publish the peer review history of their article (what does this mean?). If published, this will include your full peer review and any attached files.

Reviewer #1: No

Reviewer #2: No

---

## [Editor Report · Acceptance letter]

6 Jan 2021

PONE-D-20-19784R1 

Comprehensive Live-cell Imaging Analysis of Cryptotanshinone and Synergistic Drug-Screening Effects in Various Human and Canine Cancer Cell Lines 

Dear Dr. Wilson-Robles:

I'm pleased to inform you that your manuscript has been deemed suitable for publication in PLOS ONE. Congratulations! Your manuscript is now with our production department. 

Kind regards, 

on behalf of

Dr. Chakrabhavi Dhananjaya Mohan 

Academic Editor

PLOS ONE